TECHNIQUES AND RESOURCES

# Application and measurement of tissue-scale tension in avian epithelia *in vivo* to study multiscale mechanics and inter-germ layer coupling

Panagiotis Oikonomou*, Lisa Calvary*, Helena C. Cirne, Andreas E. Welch, John F. Durel, Olivia Powell, Kwantae Kim and Nandan L. Nerurkar‡

## ABSTRACT

As cross-disciplinary approaches drawing from mechanics have increasingly influenced our understanding of morphogenesis, tools to measure and perturb physical aspects of embryogenesis have expanded as well. However, it remains a challenge to measure mechanical properties and apply exogenous tissue-scale forces *in vivo*, particularly for epithelia. Exploiting size and accessibility of the chick embryo, we describe a technique to apply and measure exogenous forces in the order of ∼1-100 μN to the endoderm. To demonstrate the utility of this approach, we performed several proof-of-concept experiments, revealing fundamental, yet unexpected, mechanical behaviors in the early embryo. This included heterogeneous single-cell mechanotypes within the endoderm, a complex non-cell autonomous mechanical role for actin, and tight mechanical coupling across germ layers. To illustrate the broader utility of this method, we expanded this approach to the ectoderm as well, where the mechanical behavior of neural plate cells was distinct from that of the endoderm. These findings provide basic insights into the mechanics of embryonic epithelia *in vivo* in the early avian embryo, and provide a useful tool for future investigations of how morphogenesis is influenced by mechanical factors.

KEY WORDS: Epithelial mechanics, Chick, Endoderm, Neurulation, Cell morphometry

## INTRODUCTION

Development requires coordination of cell fate specification with mechanical forces that deform embryonic tissues into precise shapes. Accordingly, a mechanically motivated framework for studying development has provided increasingly cross-disciplinary insights, particularly over the past two decades. This physical view of development has revealed that mechanisms of morphogenesis can be understood using purely mechanical descriptions, providing insights even without directly implicating genes or pathways (Savin et al., 2011; Shyer et al., 2013; Tallinen et al., 2014; Mongera et al., 2018; Saadaoui et al., 2020; Serra et al., 2023). At the same time,

Department of Biomedical Engineering, Columbia University, New York, NY 10027, USA.
*These authors contributed equally to this work

‡Author for correspondence (nln2113@columbia.edu)

P.O., 0000-0002-6312-361X; N.L.N., 0000-0003-1309-8919

forces and mechanical properties are increasingly recognized as guiding cell behaviors as well (Farge, 2003; Barriga et al., 2018). Therefore, mechanics has increasingly become woven into our understanding of morphogenesis both as an effector and a modifier of developmental programs (Collinet and Lecuit, 2021). This has led to development of a range of tools for measuring mechanics of embryonic tissue *in vivo* (Gómez-González et al., 2020; Mongera et al., 2018; Chan et al., 2023; Campàs et al., 2014; Dzementsei et al., 2022) and in explants (Savin et al., 2011; Zamir et al., 2003; Barriga et al., 2018; Shook et al., 2018; Zhou et al., 2009), as reviewed by Loffet et al. (2023). However, these efforts have primarily focused on mesenchymal tissues, or whole embryos (Kunz et al., 2023; Nelemans et al., 2017 preprint). Although methods have been successfully developed to investigate epithelial mechanics *in vitro* (Harris et al., 2012; Safa et al., 2024 preprint), it remains a challenge to simultaneously apply and measure tissue-scale forces in epithelia *in vivo* (Doubrovinski et al., 2017). As a result, mechanical behaviors of embryonic epithelia remain under-explored. Here, we describe a method to apply exogenous, uniaxial stretch to epithelia in the chick embryo *in vivo*. Focusing primarily on the presumptive midgut endoderm, we conducted a series of experiments investigating mechanical heterogeneity at both tissue and cell-length scales, and quantifying mechanical coupling between endoderm and mesoderm. Finally, to illustrate broader utility of the method, we applied tension to the ectoderm, quantifying the tensile forces necessary to unzip the forming neural tube.

## RESULTS AND DISCUSSION

### *In vivo* application and measurement of tension to intact epithelia in the chick embryo

To study mechanical properties of the endoderm *in vivo*, we devised a simple approach for applying tension directly onto the flat endodermal epithelium of the presumptive chick midgut. We designed an apparatus that sits on a microscope stage, comprising a linear actuator driving bidirectional displacement of a pair of fine tungsten cantilevers to apply tension to the embryo (Fig. 1A). Cantilever bending is used to calculate the associated force (Fig. S1A). Cantilevers of 0.13 mm diameter and 40 mm length were used to measure forces in the range of 1-1000 μN (equivalent to the weight of 100 nl to 100 μl of water, respectively). To transmit displacement of tungsten rods to endoderm stretch, we used an approach motivated by the EC culture method (Chapman et al., 2001), placing small L-shaped pieces of filter paper along the lateral edges of the endoderm of Hamburger Hamilton (HH) (Hamburger and Hamilton, 1951) stage 13 embryos (Fig. 1B-D). Filter paper adhered tightly to the ventral epithelium, such that when stretch was applied via bilateral displacement of the tungsten rods, large 2D deformations could be directly observed in embryos after focal electroporation of the

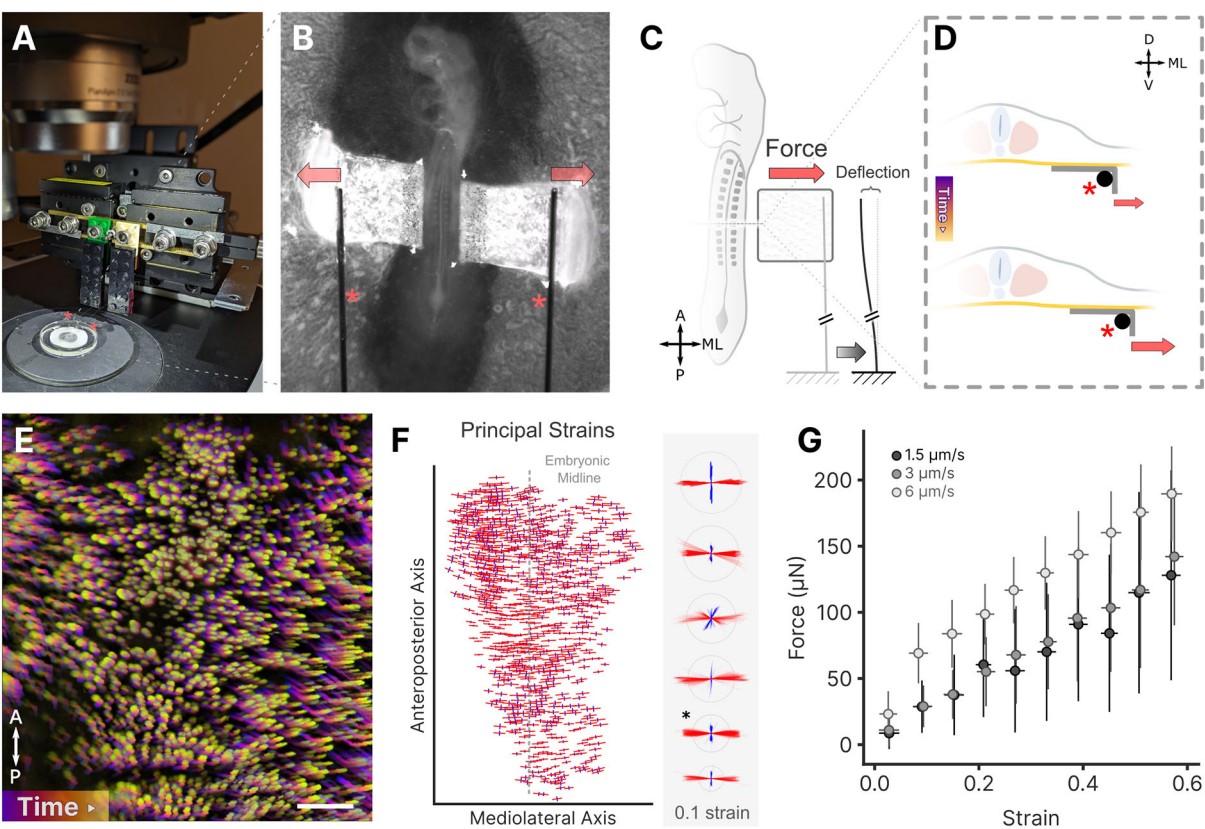

**Fig. 1. Overview of apparatus and sample stretching.** (A) Stretching apparatus sitting atop the microscope stage. Red asterisks denote the tungsten cantilevers. (B) Magnified view of cantilevers bilaterally engaging 'L-shaped' filter papers adherent to the endoderm. Asterisks denote the tungsten cantilevers and arrows indicate their relative motion. (C) Schematic of simultaneous stretching and force measurement via measurement of cantilever deflection. A, anterior; D, dorsal; ML, mediolateral; P, posterior; V, ventral. (D) Schematic transverse representation of the 'L-shaped' filter paper attachment to endoderm and engaging the cantilever, which is displaced laterally to stretch the endoderm. (E) Color-coded time projection of applied deformation in endoderm following electroporation with pCAG-H2B-GFP to visualize endoderm cell nuclei. Scale bar: 100 µm. (F) Left: Principal Lagrangian strains for a representative embryo under exogenous mediolateral stretch; length and orientation of red/blue lines indicates the magnitude and orientation maximum/ minimum principal strains. Right: Superposition of maximum (red) and minimum (blue) principal strains for six embryos; dashed circle indicates strain of 0.1; asterisk marks the representative embryo shown on the left. (G) Force versus tissue strain at 1.5 µm/s, 3 µm/s and 6 µm/s; there was no significant difference in stiffness with displacement rate. Circles represent the mean, error bars represent s.d.

endoderm with ubiquitously expressed pCAG-H2B-EGFP to visualize cell nuclei (Nerurkar et al., 2019) (Fig. 1E). 2D Lagrangian strains were largely homogeneous, with antero-posterior shortening (negative strains; Fig. 1F, blue) accompanying elongation (positive; Fig. 1F, red) along the direction of applied stretch, the mediolateral axis. Strains were similar between gut-forming endoderm and hypoblast (Fig. S1B), suggesting similar mechanical properties despite their disparate embryonic origins and gene expression profiles (Yasunaga et al., 2005). Alignment of the directions of principal strains with the mediolateral and anteroposterior axes of the embryo indicate that a simple biaxial deformation was achieved, with little to no shear (Fig. 1F). Quantification of force associated with tissue deformations revealed a linear relationship between the two, with an effective endoderm stiffness of approximately 100 µN/unit Lagrangian strain (Fig. 1G). The endoderm was highly extensible, accommodating 50-70% stretch before rupturing.

### Identification of single-cell mechanotypes through application of exogenous stretch

Applying exogenous stretch to the endoderm *in vivo* provides access to many questions regarding the mechanical behaviors of embryonic epithelia. To begin, we asked how tissue-scale stretch is translated locally to cell deformations, relying on electroporation of pCAG-EGFP-CAAX to visualize endoderm cell boundaries (Fig. 2A, Movie 1). Automated cell segmentation (Stringer et al., 2021) and tracking (Ulicna et al., 2021) were used to quantify the evolution of single-cell morphometrics as the endoderm was progressively stretched (Fig. 2B), and used to quantify features such as cell area, aspect ratio, strain and position. To handle these large datasets (20 features for each of approximately 20,000 cells), we employed uniform manifold approximation and projection (UMAP) to create a low-dimensional representation of these high-dimensional measurements (see Materials and Methods) (Shannon et al., 2024). Across six embryos, cells were broadly distributed in the UMAP space, revealing distinct 'mechanotype', populations of cells within the same tissue with dissimilar mechanical properties (Fig. S2A). To study this heterogeneity further, we performed unsupervised clustering of the cells by their similarity in features (Fig. 2C). Clusters largely included entire cell trajectories, such that the evolution of features can be studied as a function of applied strain (Fig. S2B). The dominant structure in the UMAP followed an axis of increasing cell areas (Fig. 2D, Fig. S3A), with secondary separation according to applied strain and area changes (Fig. S2B,C). Whether this mechanotype heterogeneity is itself functionally important for gut tube morphogenesis or an indirect biophysical consequence of

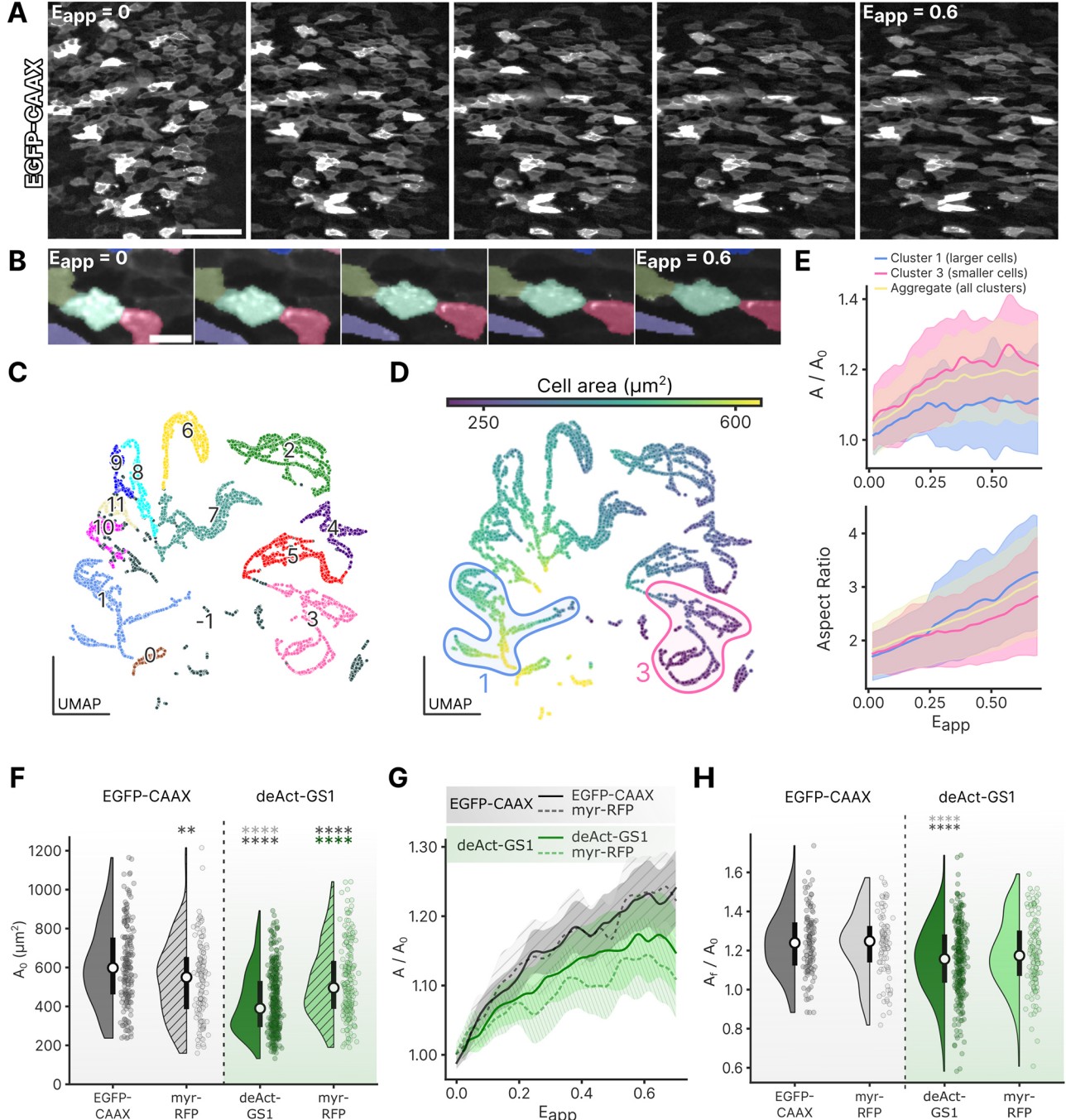

**Fig. 2. Cell response to applied strain reveals mechanotype heterogeneity.** (A) Stills from a time-lapse movie of EGFP-CAAX-expressing endoderm cells as applied strain $E_{app}$ is increased from 0 (left) to 0.6 (right). Scale bar: 100 µm. (B) Magnified view of single-cell deformation as indicated by automated segmentation and tracking as $E_{app}$ increases from 0 (left) to 0.6 (right). Scale bar: 10 µm. (C) Eleven distinct cell clusters identified in UMAP space; each data point indicates a single cell at a single time step during the application of stretch. (D) UMAP space color-coded by cell area. (E) Evolution of normalized cell area ($A/A_0$) (top) and aspect ratio (bottom) with increasing applied strain $E_{app}$ for all cells, cluster 1 (large cells) and cluster 3 (small cells), and aggregate (all clusters). Solid lines indicate mean and shaded area represents s.d. (F) Initial cell areas ($A_0$) of EGFP-CAAX, deAct-GS1 and respective wild-type (myr-RFP) neighboring cell. **$P<0.01$, ****$P<0.0001$ (one-way ANOVA with Tukey's HSD). (G) Evolution of mean normalized cell area ($A/A_0$) with increasing strain $E_{app}$ for EGFP-CAAX, deAct-GS1 and respective wild-type cells. Shaded areas represent s.d. (H) Quantification of final increase in cell area ($A_f/A_0$). ****$P<0.0001$ (one-way ANOVA with Tukey's HSD). In F and H, white circle represents the median and black bar represents the interquartile range.

biochemical differences between cell types is not yet clear. One could envision that cell rearrangements in an epithelium could be facilitated by mechanotype heterogeneity to avoid jamming. For each cluster, stretching increased cell area and aspect ratio as cells rotated toward the direction of applied stretch (Fig. S3C). However, while aspect ratio increased linearly with applied stretch, area changes were nonlinear, with cells first accommodating stretch through area expansion before plateauing (Fig. 2E, Fig. S3C). Interestingly, cells with larger cross-sectional area attenuated their area expansion at lower applied strains than smaller cells (Fig. 2E, Fig. S3B,C). This may be due to a tendency to preserve volume as cells deform (Wilkes and Athanasiou, 1996), shortening apicobasally to

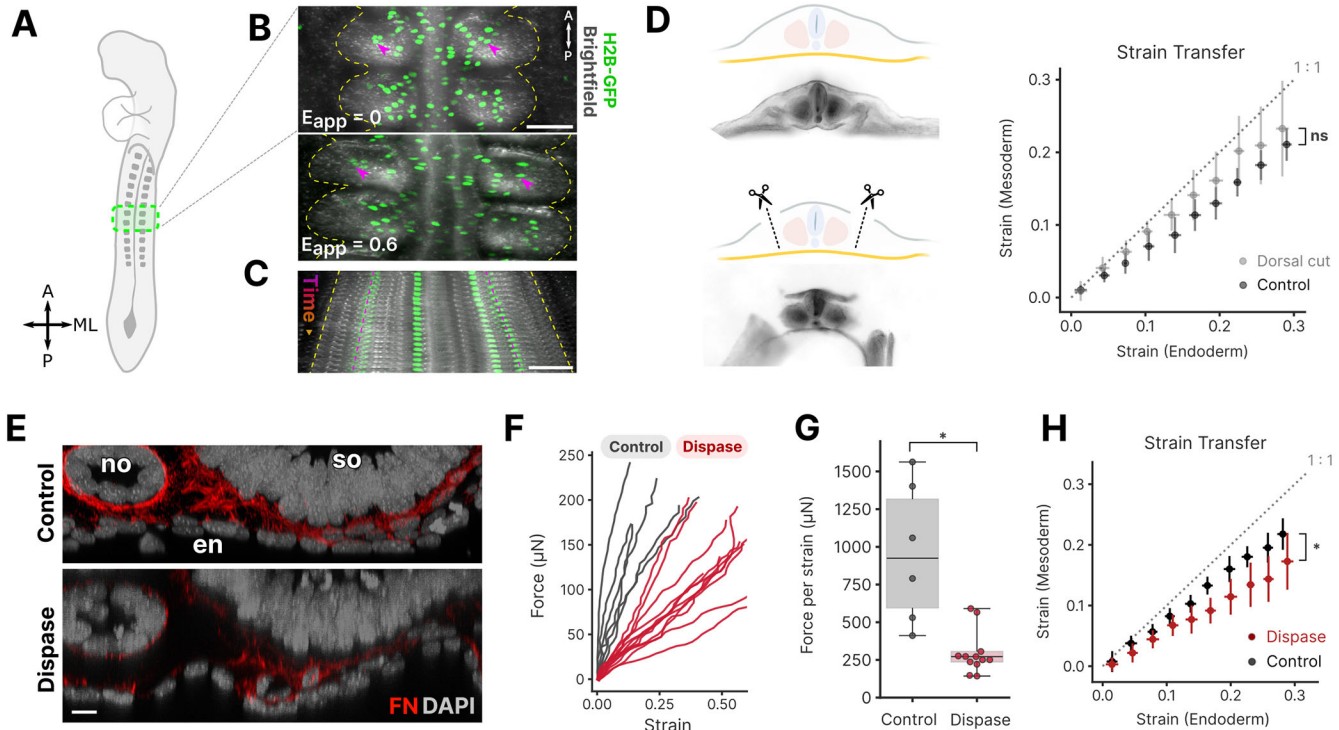

**Fig. 3. Mechanical coupling across germ layers.** (A) Schematic of the embryonic region shown in B,C. Ventral view. A, anterior; ML, mediolateral; P, posterior. (B) Overlay of brightfield and fluorescent images prior to (top; applied strain $E_{app}$=0) and after application of stretch to the endoderm (bottom; $E_{app}$=0.6); endoderm nuclei are labeled by electroporation with pCAG-H2B-GFP (green). Magenta arrowheads and dashed yellow lines denote the positions of two nuclei and lateral somite boundaries, respectively. Scale bar: 100 µm. (C) Kymograph of stretch progressively applied to endoderm, illustrating concomitant deformation of somites over time. Scale bar: 100 µm. (D) Left: Schematic (top) and brightfield images (bottom) of the embryo following dorsal cuts to isolate ventral endoderm–somite interactions. Right: Comparison of Lagrangian strains in endoderm and mesoderm for control and dorsal cut embryos. ns, not significant ($P$=0.52, Welch's $t$-test on the slopes of the strain transfer per sample). Circles represent the mean, error bars represent s.d. (E) Transverse view of fibronectin (FN; red) staining of the endoderm–somite interface of control (top) and Dispase-treated (bottom) embryos. DAPI was used to stain nuclei (gray). en, endoderm; no, notochord; so, somite. Scale bar: 10 µm. (F) Quantification of applied force versus in-plane Lagrangian strain in the endoderm for control and Dispase-treated embryos. (G) Relative stiffness quantified from the force-strain curves shown in F. *$P$=0.012 (Welch's $t$-test). (H) Comparison of Lagrangian strains in endoderm and mesoderm for control and Dispase-treated embryos. *$P$=0.012 (Welch's $t$-test on the strain transfer per sample). Circles represent the mean, error bars represent s.d.

accommodate in-plane stretch until the balance between volumetric constraints (incompressibility of intracellular contents) and boundary constraints (cell–cell and cell–matrix contacts) begins to favor cell shape changes that do not further alter the cell height and area.

## Endoderm-specific disruption of F-actin attenuates cell area expansion in response to tissue-scale stretch

With the ability to apply tension to epithelia *in vivo*, one can test the contribution of specific subcellular components through targeted disruption of their normal function. For illustration, we focused on F-actin as a key structural component dictating passive and active cell properties (Brückner et al., 2019; Bisaria et al., 2020). F-actin was disrupted in an endodermal subpopulation by focal electroporation to misexpress DeAct-GS1 (DeAct), a GFP-fused peptide that sequesters G-actin monomers, leading to F-actin depolymerization (Harterink et al., 2017) (Fig. S4A). To examine non-cell-autonomous effects of F-actin disruption, a second electroporation was carried out to visualize neighboring cells by misexpression of a membrane-localized RFP (myr-RFP) (Fig. S4B). Prior to stretching, F-actin disruption by DeAct-GS1 significantly decreased cell cross-sectional area compared to neighboring myr-RFP-expressing cells and compared to cells expressing membrane-localized GFP (EGFP-CAAX) in control electroporated embryos (Fig. 2F). We next examined how F-actin disruption influences stretch-dependent changes in cell morphometrics. In DeAct-electroporated embryos, both myr-RFP-

and DeAct-GS1-expressing cells displayed limited area changes with stretch compared to myr-RFP- and EGFP-CAAX-expressing cells in control embryos (Fig. 2G, Movie 2). This was unexpected for two reasons. First, in controls, cells with a smaller initial area experienced larger area changes under applied stretch (Fig. S3B), but smaller DeAct-GS1 did not follow this trend. Second, myr-RFP-positive cells adjacent to DeAct-GS1-expressing cells had initial cross-sectional areas similar to control cells, but responded to stretch as DeAct-expressing cells did, with reduced area changes as stretch is applied (Fig. 2H). These findings suggest that F-actin may play an active role in apicobasal shortening to accommodate in-plane stretch (Sherrard et al., 2010). Indeed, DeAct-GS1 cells had a larger initial cell height, which remained significantly elevated upon stretching (Fig. S4E-G). Alternatively, it is possible that DeAct-GS1 disrupts important force transmission through cytoskeletal connectivity, muting the Poisson effect responsible for apicobasal shortening with in-plane expansion. Nonetheless, aspect ratio varied with strain similarly across all cells (Fig. S4C). F-actin disruption also did not significantly alter relative tissue stiffness (Fig. S4D), suggesting that endoderm properties may be dominated by the stiff basement membrane rather than active cell properties. These findings support an important role for F-actin in transmitting macroscopic tension in the epithelium to local cell deformations in a non-cell-autonomous manner, and, more generally, highlight the importance of studying the mechanical response of epithelial cells *in vivo*, where complex

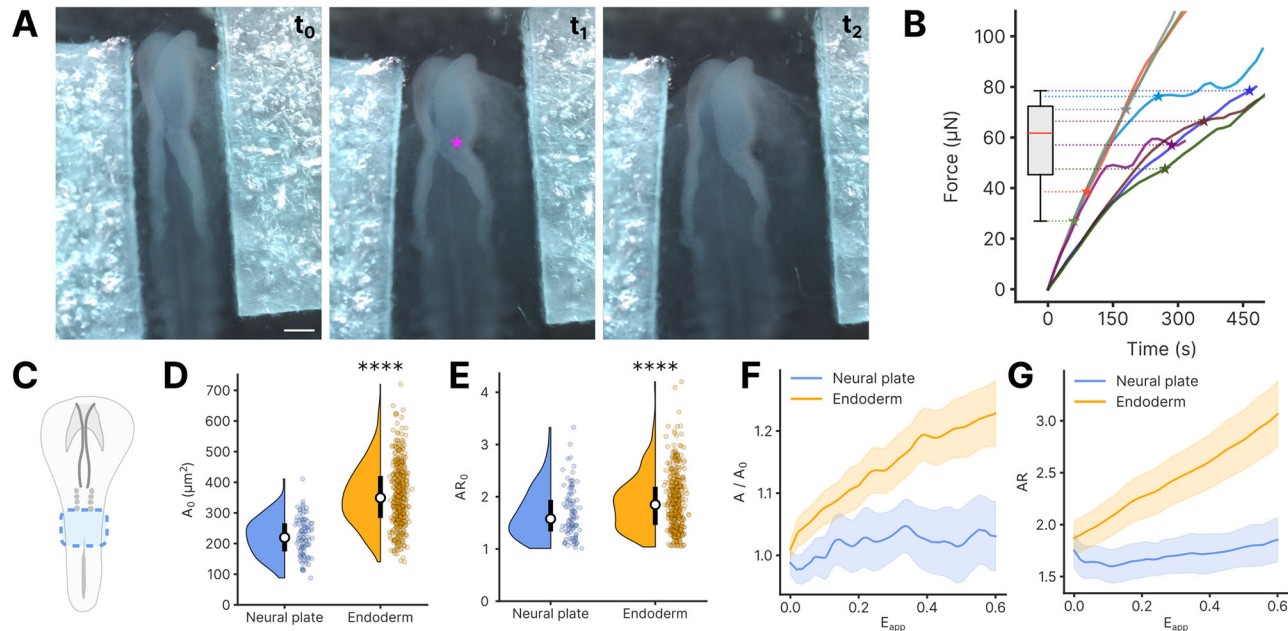

**Fig. 4. Tensile mechanics of the ectoderm.** (A) Snapshots of an HH stage 8 embryo as tension is applied across the surface ectoderm; magenta star indicates the rupture point of dorsal neural tube. Scale bar: 200 µm. (B) Measurement of the applied force as a function of time. Stars mark the rupture force per sample ($n$=8), each indicated by a different color. (C) Schematic of the embryonic region (neural plate, HH stage 8) stretched in D-G. (D,E) Initial cell areas ($A_0$, D) and aspect ratios ($AR_0$, E) of neural plate and endoderm cells. ****$P$<0.0001 (Student's $t$-test). (E) Initial aspect ratio of neural plate and endoderm cells. ****$P$<0.0001 (Student's $t$-test). In D and E, white circle represents the median and black bar represents the interquartile range. (F) Evolution of mean normalized cell area ($A/A_0$) with increasing applied strain$E_{app}$. Solid lines indicate mean and shaded area represents s.d. (G) Evolution of mean aspect ratios ($AR$) with increasing applied strain$E_{app}$. Solid lines indicate mean and shaded area represents s.d. Endoderm data reproduced from Fig. 2 for comparison.

interactions of cells with their neighbors and extracellular matrix (ECM) may produce counter-intuitive results.

## Exogenous stretch reveals mechanical coupling between endoderm and mesoderm

While investigating the effects of tension on in-plane properties of the endoderm, we observed that as the endoderm is stretched as much as 75% of the applied deformation was transmitted to the sub-adjacent somites (Fig. 3A-C, Movie 3), indicating potential mechanical coupling between germ layers over large deformations and forces in the order of ~100 µN. To test whether mesodermal deformations result from direct mechanical coupling to the endoderm, or indirect effects caused by lateral fusion of dorsal and ventral extra-embryonic tissues that may compress the somites as the embryo is stretched, we examined strain transfer between germ layers following microdissection to dorsally isolate the embryo proper from extra-embryonic tissues (Fig. 3D). Following dissection, strain transfer between endoderm and somites was unaffected (Fig. 3D), suggesting that this phenomenon results directly from mechanical coupling across the interface between endoderm and somites. Similar strain transfer was observed posteriorly from endoderm to presomitic mesoderm as well (Fig. S5).

Based on previous identification of fibronectin pillars that span the space between the endoderm and somites (Sato et al., 2017), we next considered whether such ECM components could be responsible for mechanical coupling. To disrupt fibronectin and collagens in the ECM, Dispase treatment (Zaher et al., 2025) was used to disrupt the fibronectin-rich ECM that forms the interface between endoderm and paraxial mesoderm (Fig. 3E). This resulted in an 80% reduction in the in-plane stiffness of the endoderm (Fig. 3F,G). Surprisingly, however, only a modest reduction in strain

transfer between endoderm and mesoderm was observed (Fig. 3H). This may be because fibronectin was only partially removed (Fig. 3E), or that Dispase-sensitive ECM components play a more dramatic role in determining the in-plane endoderm properties than they do in mechanical coupling with mesoderm.

In light of this coupling, it is surprising that exogenous stretch produces largely homogeneous strain fields throughout the midgut endoderm (Fig. 1G, Fig. S1B), despite the varied architecture and cell types of neighboring mesodermal tissues, including splanchnic, intermediate, axial and paraxial mesoderm. This could potentially be explained if the endoderm is significantly stiffer than the mesoderm. The tight mechanical coupling between presomitic mesoderm and a stiffer endodermal layer may have implications for a range of events, such as axis elongation and gut tube morphogenesis, which have traditionally been studied as mechanically distinct.

Although mechanical interactions between germ layers have been implicated in many aspects of vertebrate morphogenesis (Smith et al., 2022), including placement of the neural anlage (Smutny et al., 2017), neurulation (Guillon et al., 2020), and axis elongation (Xiong et al., 2020), in most cases, these interactions have been inferred from mathematical modeling (Guillon et al., 2020), or indirectly by observing how tissue compartments respond to perturbations in the neighboring tissue (Xiong et al., 2020; Smutny et al., 2017). By providing access to directly assess interfacial mechanics of embryonic tissues *in vivo*, the present approach may therefore be useful in addressing a range of problems in morphogenesis.

## Measurement of tensile properties of the neuroepithelium during primary neurulation

To illustrate the broader utility of this method, we extended this technique to investigate ectodermal tissues as well. We first

DEVELOPMENT

measured the forces needed to unzip the neural tube during primary neurulation, finding that rupture of the posterior-most adhesion point in the neural tube was achieved with forces in the order of ~50 µN (Fig. 4A,B, Movie 4). Next, we investigated whether ectodermally derived epithelia respond to stretch similarly to the endoderm, focusing on the relatively flat posterior neural plate at HH stage 8 (Fig. 4C). Comparing to endoderm, we found that neural plate cells initially have smaller cross-sectional area and are less elongated (Fig. 4D,E, Fig. S6A). With exogenous stretch, neural plate cells underwent minimal changes in area (Fig. 4F, Fig. S6D) or aspect ratio (Fig. 4G, Fig. S6E). These data suggest that, in contrast to endoderm, neural plate may accommodate stretch primarily through intercellular deformations, with limited force transmission across cells. Apparent tissue stiffness, however, was similar to that of endoderm (Fig. S6B,C). Overall, these findings illustrate the potentially broad applications of a relatively simple, inexpensive technique for measuring tension while applying stretch to embryonic epithelia in vivo, and suggest that mechanics of cell response to tension vary within and across epithelia in the early chick embryo.

## MATERIALS AND METHODS
### Chicken embryology techniques, electroporation and drug treatments
Fertilized White Leghorn chicken (*Gallus gallus domesticus*) eggs were acquired from University of Connecticut Poultry Farm and incubated at 37°C and 60% humidity for 45 h (HH stage 12). Embryos were harvested onto filter paper rings and transferred to plates containing EC culture medium (Chapman et al., 2001). Endoderm-specific electroporation was carried out as previously described (Nerurkar et al., 2019; Oikonomou et al., 2023). Briefly, electroporation solutions were prepared by diluting pCAG-H2B-EGFP (Addgene plasmid #32599), pCAG-EGFP-CAAX (Addgene plasmid #86056), pCAG-myr-mRFP (Addgene plasmid #32604), or pCAG-DeAct-GS1 (a fusion protein of DeAct-GS1 with EGFP, Addgene plasmid #89445) plasmids to 3.5 µg/µl in molecular grade ddH2O with 5% sucrose and 0.1% Fast Green FCF. Electroporation solution for neural plate epithelium electroporation was prepared by diluting pCAG-EGFP-CAAX to 1 µg/µl in molecular grade ddH2O with 0.1% Fast Green FCF. DeAct-GS1 contains the gelsolin fragment known as segment 1, which binds to actin monomers and renders them unable to polymerize into F-actin. Similar to pharmacological sequestering of actin monomers (Ayscough et al., 1997), this results in destabilization and pronounced depolymerization of F-actin by disequilibrium of filamentous and monomeric actin in the cytoplasm (Harterink et al., 2017). More intense disruption of F-actin using the SpvB variant of DeAct (Harterink et al., 2017) resulted in delamination of cells (not shown), and we therefore focused instead on DeAct-GS1, which has a more moderate effect, for further study. Following delivery of the DNA solution to the ventral surface by micropipette, embryos were electroporated using a Nepa 21 transfection system (Nepa Gene) with a sequence consisting of three 35 V poring pulses of 0.2 ms duration separated by 50 ms with a decay rate of 10% between successive pulses, followed by five 4 V transfer pulses of 5.0 ms duration separated by 50.0 ms with a 40% decay rate (Nerurkar et al., 2019; Oikonomou et al., 2023). Experiments were carried out after 6-8 h incubation *ex ovo*, when embryos had reached HH stage 13 and fluorescent signal was readily detected. For the dorsal ectoderm dissections, embryos were placed dorsal side up under a Zeiss Stemi 508 stereo microscope. Using a flame-sharpened fine tungsten knife, bilaterally symmetric incisions were made to sever the surface ectoderm at the level of the posterior-most four or five somites. Embryos were then imaged as described below. To quantify strain in the presomitic mesoderm, tissues were labeled by microinjection of the lipophilic dye DiI (2.5 mg/ml in N,N-dimethylformamide, Invitrogen) using pulled glass capillary needles (Fig. S5A). For Dispase treatment, embryos were fitted with a small plastic confining ring 1 cm in diameter to retain Dispase solution, and treated with 100 µl of a 1 U/ml Dispase (Dispase I protease, Sigma-Aldrich, D4818-2MG) solution diluted in PBS; vehicle controls similarly received 100 µl of

PBS. Embryos were incubated for 10 min at room temperature, then rinsed in PBS, and excess moisture was wicked using a Kimwipe prior to stretching experiments. Mild conditions were used to sever the ECM without completely stripping it from the tissue, as more complete Dispase digestion results in severe delamination of the chick embryo (Danesin et al., 2021). For neural plate epithelium stretching experiments, electroporation solution was delivered in between the vitelline membrane and the neuroepithelium and experiments were carried out after 4-6 h incubation *ex ovo*. The vitelline membrane was removed just before stretching to allow the placement of the filter paper on the neural plate epithelium.

### Mechanical testing apparatus
Mechanical properties of embryonic tissues were probed using a custom micromechanical apparatus to perform uniaxial, bidirectional tensile tests. The device relies on thin tungsten rods to transmit forces to the embryo, enabling both the application of tissue strain and the measurement of the applied force (which is proportional to rod deflection; Fig. 1C, Fig. S1A). A gear system was used to achieve opposing/bidirectional displacement of a pair of tungsten rods from a single linear actuator. Each tungsten rod was fixed to a spring-loaded linear stage (Newport, 460A-X), and the two stages were in turn mounted with opposing orientations on a base plate (Newport, M-BK-3A-T). One stage was directly controlled using a linear actuator (Newport, CONEX-TRA25CC), and this movement was coupled to opposing movement of the second stage using a gear system in which a centrally positioned gear (McMaster-Carr, 7880K14) rotates freely on a perpendicular rotary shaft (McMaster-Carr, 1327K93) to convert rightward motion from a gear rack (McMaster-Carr, 7854K11) mounted on the actuated stage into leftward motion on another gear rack that is fastened to the passive stage. On each stage, a tungsten rod of 0.127 mm diameter (A-M Systems, 716000) was attached to act as the cantilever arm. Finally, stretching of the endoderm was achieved by movement of the tungsten cantilevers against filter paper squares (2×2 mm) placed directly on the ventral surface of the embryo; similar experiments on the neural tube were conducted by dissecting away the vitelline membrane and placing filter paper squares on the surface ectoderm to the left and right of the forming neural tube. While smaller filter paper pieces could be used to provide more focal tensile forces, a larger size was used in the present study to avoid confounding boundary effects, ensuring that observed cell mechanotype heterogeneity could be attributed to intrinsic mechanical differences of the endoderm rather than artificial effects of large stress gradients at the filter paper edges. Of note, changing the filter paper size will change the force distribution, and smaller filter papers may require use of cantilevers with lower bending stiffness (e.g. longer, thinner beams) to avoid stress concentrations that damage the tissue.

Bending stiffness of the tungsten cantilever was calibrated by measuring vertical deflection of the cantilever tip after hanging small pieces of aluminium foil of with known weights from the cantilever tip. Bending stiffness (0.167 N/m) was then quantified as the slope of the force versus deflection data, as quantified by a linear regression (Fig. S1A). Empty runs in which the cantilever was displaced at a constant rate without load was used to test variation in true versus expected tip position, concluding that error increased over time but did not exceed 6% (Fig. S1C). The tungsten cantilevers were displaced slowly at a rate of 3 µm/s to mitigate viscous effects; no differences were observed when comparing tensile response across a range of displacement rates, confirming minimization of viscous effects (Fig. 1G). Cantilever-based force measurements can be highly versatile: bending stiffness varies with beam radius $R^4$ and inversely with length $L^3$, such that bending stiffness can be tuned across a wide range of magnitudes, including 1-10 nN during zebrafish gastrulation (Krieg et al., 2008) and cardiac c-looping in the chick (Zamir et al., 2003). Force estimates using other techniques suggest that this range is appropriate to investigations such as avian axis elongation (Chan et al., 2023), gastrulation and convergent extension in *Xenopus* (Shook et al., 2018), and epithelial morphogenesis in zebrafish (Yamada et al., 2017), among others. Replacing the filter papers in the present set-up with solid platens (Shyer et al., 2013), or micro-indenters (Zamir et al., 2003; Nerurkar et al., 2006), allows the same approach to be modified to apply compression instead of tension.

TECHNIQUES AND RESOURCES

Therefore, the range of force magnitudes accessible with this approach could extend its utility to a range of other applications. However, it is worth noting that the tight coupling between filter paper and chick tissue in the present study is essential for force transmission to the tissue, and this would have to be fine-tuned for each tissue or organism of interest. Nonetheless, it is likely that this technique will be useful in other developmental model organisms, as well as for studying the mechanobiology of tissue explants and organoids.

Live imaging of samples was performed as stretch was progressively applied using a ZEISS Axio Zoom.V16, at 50× magnification to visualize displacement of cell nuclei, and at 100× to visualize cell shape changes. Images were captured every 15-20 s for a total of 10-15 min or until tissue rupture. Imposed deformations were sufficiently large to warrant use of Lagrangian strains to quantify the deformation field, as opposed to linear/engineering strains. Accordingly, the tissue-scale applied strain was quantified as $E_{app} = \frac{1}{2}(\lambda^2 - 1)$, where $\lambda = L/L_0$ is the stretch ratio relating the current/deformed distance between fiducial markers near the two filter paper squares (e.g. labeled cells) normalized to their distance prior to the onset of stretch. To report the local deformations within the epithelial plane (Fig. 1G), the full Lagrangian strain tensor was computed as $\mathbf{E} = \frac{1}{2}(\mathbf{F}^T\mathbf{F} - \mathbf{I})$, where $\mathbf{F} = \frac{\partial \mathbf{x}}{\partial \mathbf{X}}$ is the deformation gradient tensor and $\mathbf{x}$ and $\mathbf{X}$ are position vectors in the deformed and undeformed configuration, respectively. $\mathbf{F}$, and subsequently $\mathbf{E}$, were computed following the approach of Mowlavi et al. (2022), using as input the trajectories of all cells tracked in the tissue with btrack (Ulicna et al., 2021). The direction and magnitude of principal strains were calculated by eigen-decomposition of the 2D strain tensor.

For cell height quantification of EGFP-CAAX, myr-RFP and deAct-GS1 electroporated endoderm cells following application of exogenous stretch, tension was applied manually using a custom 3D-printed apparatus to displace filter paper patches, enabling fixation in the pre-stretched state to examine effects of in-plane tension on cell height. Height was imaged by fluorescence confocal microscopy as described below.

Additional instructions to assemble the stretching apparatus have been deposited in the online repository, along with all custom code used for strain maps and other data quantification (see https://github.com/eigenP/OikonomouCalvaryNerurkar2024/blob/main/Straindoderm%20Device%20Assembly%20and%20Usage%20Guide.pdf).

### Immunofluorescence and staining
Embryos were fixed overnight in 4% paraformaldehyde in PBS, rinsed, and dissected from surrounding extra-embryonic tissue and filter paper. Embryos were washed (PBS with 0.1% Triton X-100 was used for all washes) and left in blocking solution (10% heat inactivated goat serum) for 2 h, then incubated overnight at 4°C with mouse anti-fibronectin primary antibody (DSHB, B3/D6) diluted in PBS+0.1% Triton X-100 at 1:200. Samples were washed and then incubated with a goat anti-mouse 488 secondary antibody (1:200, diluted in PBS+0.1% Triton X-100) and DAPI (1:1000) overnight. Embryos were then cleared in RapiClear (RapiClear 1.49, Sunjin Labs), and imaged on a ZEISS LSM880 confocal microscope with a 40× water immersion objective, and a dense z-stack at 0.2 µm spacing for optical section reconstruction. For F-actin staining, embryos were incubated 2 h with Phalloidin-iFluor 647 (Abcam, ab176759; 1:500) in PBS+0.1% Triton X-100. Embryo were imaged on a ZEISS LSM880 confocal microscope with a 63× oil immersion objective with 0.3 µm spacing for optical section reconstruction. Representative images were denoised using the difference of Gaussians tool from the CLIJ2 ImageJ plugin (Sigma1x=1; Sigma2x=20; Sigma1y=1; Sigma2y=20).

### Image analysis and data processing
The code to follow the analysis pipeline is available at https://github.com/eigenP/OikonomouCalvaryNerurkar2024/. Image analyses on time-series movies obtained from the mechanical testing experiments were performed using the open-source image visualization Python library Napari (Sofroniew et al., 2022), preprocessing functions from the sci-kit image library (van der Walt et al., 2014), the segmentation algorithm Cellpose (Stringer et al., 2021) (starting from weights of the cyto2 model, trained with manual annotations for 1000 epochs on a workstation equipped with an NVIDIA T400 GPU), and btrack, a Python library for multi-object Bayesian tracking

(Ulicna et al., 2021) (particle configuration). Where necessary, time-lapse movies were stabilized against drift using custom code as described by Oikonomou et al. (2023), inspired by Fast4DReg (Pylvänäinen et al., 2023). Cell morphometry features generated through this pipeline were aggregated across biological replicates and plotted in UMAP space, generally following the approach of cellPLATO (Shannon et al., 2024) and adapting code from ColabTracks (Jacquemet, 2024 preprint). Specifically, UMAP embedding was based on the following features: magnitude of applied strain, deformed aspect ratio, undeformed aspect ratio, current cell area, undeformed cell area, orientation of the cell major axis relative to the mediolateral embryonic axis, minor axis length, major axis length, cell perimeter, and shape solidity (area/convex area). Clustering was performed on the UMAP-transformed data using HDBSCAN (McInnes et al., 2017).

### Acknowledgements
We thank members of the Nerurkar lab for their valuable scientific input. We also thank Joseph Viola for help with designing and prototyping the mechanical stretching apparatus, and the image.sc forum users for their advice on image analysis tasks.

### Competing interests
The authors declare no competing or financial interests.

### Author contributions
Conceptualization: P.O., L.C., N.L.N.; Data curation: P.O., L.C., A.E.W.; Formal analysis: P.O., L.C., H.C.C., N.L.N.; Funding acquisition: N.L.N.; Investigation: P.O., L.C., H.C.C., A.E.W., J.F.D., N.L.N.; Methodology: P.O., L.C., H.C.C., A.E.W.; Resources: P.O., L.C., O.P., K.K., N.L.N.; Software: P.O., L.C.; Supervision: N.L.N.; Validation: P.O.; Visualization: P.O., L.C.; Writing – original draft: P.O., L.C., N.L.N.; Writing – review & editing: P.O., L.C., N.L.N..

### Funding
This work was funded by the National Institute of General Medical Sciences (R35GM142995 to N.L.N.). Additional support was provided from the Columbia University Digestive and Liver Disease Research Center (1P30DK132710). Open Access funding provided by Columbia University. Deposited in PMC for immediate release.

### Data and resource availability
Code to replicate the analysis can be found at https://github.com/eigenP/OikonomouCalvaryNerurkar2024/ (in the form of Python Jupyter notebooks). Representative raw image files for stretching experiments across conditions have been deposited in Figshare under the following DOIs: https://doi.org/10.6084/m9.figshare.28836638.v1 and https://doi.org/10.6084/m9.figshare.28836713.v1. All other relevant data and details of resources can be found within the article and its supplementary information.

### Peer review history
The peer review history is available online at https://journals.biologists.com/dev/lookup/doi/10.1242/dev.204561.reviewer-comments.pdf

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
