## [Peer Review File · Development (Cambridge, England)]

Application and measurement of tissue-scale tension to avian epithelia in vivo to study multiscale mechanics and inter-germ layer coupling

Panagiotis Oikonomou, Lisa Calvary, Helena Campos Cirne, Andreas Emerson Welch, John Franics Durel, Olivia Powell, Kwantae Kim and Nandan Nerurkar
DOI: 10.1242/dev.204561

Editor: James M Wells

Review timeline

Original submission:	25 November 2024
Editorial decision:	4 January 2025
First revision received:	8 May 2025
Editorial decision:	13 June 2025
Second revision received:	21 July 2025
Accepted:	4 August 2025

Original submission

First decision letter

MS ID#: dev.204561

MS TITLE: Application of tissue-scale tension to avian epithelia in vivo to study multiscale mechanical properties and inter-germ layer coupling

AUTHORS: Panagiotis Oikonomou; Lisa Calvary; Helena Campos Cirne; Andreas Emerson Welch; John Franics Durel; Olivia Powell; Kwantae Kim; Nandan Nerurkar

Dear Dr Nerurkar,

I have now received all the referees' reports on the above manuscript, and have reached a decision. The referees' comments are appended below, or you can access them online: please go to:

As you will see, the referees express considerable interest in your work, but have some significant criticisms and recommend a substantial revision of your manuscript before we can consider publication. If you are able to revise the manuscript along the lines suggested, which may involve further experiments, I will be happy receive a revised version of the manuscript. Your revised paper will be re-reviewed by one or more of the original referees, and acceptance of your manuscript will depend on your addressing satisfactorily the reviewers' major concerns. Please also note that Development will normally permit only one round of major revision. If it would be helpful, you are welcome to contact us to discuss your revision in greater detail. Please send us a point-by-point response indicating your plans for addressing the referees' comments, and we will look over this and provide further guidance.

Please attend to all of the reviewers' comments and ensure that you clearly highlight all changes made in the revised manuscript. Please avoid using 'Tracked changes' in Word files as these are lost in PDF conversion. I should be grateful if you would also provide a point-by-point response detailing how you have dealt with the points raised by the reviewers in the 'Response to Reviewers' box. If you do not agree with any of their criticisms or suggestions please explain clearly why this is so.

Reviewer 1

SUMMARY OF THE ADVANCE MADE IN THIS PAPER AND ITS POTENTIAL SIGNIFICANCE TO THE FIELD

Oikonomou et al. presents a stretching device that can be calibrated to measure applied tissue forces in the chick embryo. The device is a uniaxial stretcher with two tungsten rods where the chick embryo is attached ex-vivo (EC-culture). The deformation of the tungsten rods can be calibrated to measure transmitted to the embryo.

To illustrate the use of this device the authors explore three different applications: 1) to analyse the effects of stretching of the endoderm at the cell level, 2) to study the mechanical coupling of the endoderm and paraxial mesoderm and 3) to measure the forces necessary to "unzip" the neural tube.

SUGGESTIONS TO AUTHORS

Main comments

- 1) The introduction lacks a series of important references on using stretching devices to characterise deformations in epithelia (e.g. Harris et al. 2012; Yang et al. 2024 - Charras & Davidson labs) and deformation in whole chicken embryos (e.g. Nelemans. et al. 2020; Kunz, et al. 2023 - Smit and Xiong labs).
- 2) Uniaxial stretching of epithelial tissues or full chicken embryos is not particularly novel. The novelty might be in the measurement of the applied forces using the calibrated cantilevers. However, the authors should stress this aspect more in the first study case: When the authors study the stretching of the endoderm, forces are only mentioned in Fig. S4. Instead, the authors do an in-depth characterisation of the response of endodermal cells to deformation and the role of f-actin. Although some of these results are interesting, they feel too long and out of place since do not contribute to showcase the ability of measuring forces.
- 3) The second (mechanical coupling of the endoderm and paraxial mesoderm) and third ("unzipping" the neural tube) cases are better representations of the strengths of the proposed method, but the third case feels incomplete in comparison to the first case where great characterisation is presented.
- 4) This work is presented as a technique and resources report. However, in my opinion to be used as a resource the authors should include a supplement with detailed information of how to build the device step by step with images and diagrams (including consideration such as the microscope...), how to calibrate it and possibly any code developed to compute the strain tensors.

Other comments:

- * As a resource paper it would benefit for a more in-depth exploration of the sources of variability in the force measurements.
- * As a resource paper it would benefit from comparison with other methods to measure tissue forces. How easy are to implement? What range of forces can measure?...
- * The mechanotypes identified in the first case of study are interesting but the biological significance and reproducibility of these UMAP clusters are not fully explored. Are they specific spatial distribution or localisations of these mechanotypes? What is the potential functional relevance of having different mechanical responses within the same epithelium? The fact that the main determinant of the cellular response to deformation is the original area, might suggest that these mechanotypes are composed of cells at different phases of the cell cycle. These could be accounted for and potentially regressed out of the data in future studies.
- * What is the source of mechanical coupling between the endoderm and paraxial mesoderm? Does this coupling have a functional role?

Reviewer 2

SUMMARY OF THE ADVANCE MADE IN THIS PAPER AND ITS POTENTIAL SIGNIFICANCE TO THE FIELD

This paper describes an experimental system for applying tension to chick embryos and uses the system to measure the mechanical and cellular properties of epithelia. In the system filter papers are attached to the borders of the flat chick embryo and pulling force is applied via calibrated tungsten wires. Embryonic cells are labeled fluorescently and experiments are conducted under live

imaging. Thus the strain on the epithelial cells can be correlated with the applied stretching force. The researchers use this system to make some interesting observations regarding the response of the endodermal layer to stretching, on the linkage between endodermal and mesodermal cell layers. They also describe some preliminary applications of the system to ectodermal tissue behavior with respect to neural tube closure.

The coupling of calibrated forces with live imaging at the cellular level is a powerful one and has the potential to provide new information to the study of mechanical properties of early chick embryos, particularly the external endodermal and ectodermal layers. It should be useful to chick embryologists and also to those working in other systems provided that the embryos could be adapted to experimental setup. The documentation of mechanical linkage between endoderm and mesoderm is important and is an interesting area for future study. Some suggestions are given for strengthening the paper so that it will be useful for its intended audience, particularly regarding description of the technique and clarification of embryological mechanisms.

SUGGESTIONS TO AUTHORS

1. This is a Techniques and Resources paper. Thus it is important that the Methods be described sufficiently so that others could construct the apparatus in their own labs. It would be very helpful if it could be better described how the tungsten cantilevers are attached to the filter paper, as this was not clear from the description in the paper. It would also be helpful to include a technical sketch of the system.

2. Does the total area of the filter paper and its area of contact with the embryo affect the measurements being made?

3. It is stated: "The tungsten cantilevers were displaced slowly at a rate of 0.003 mm/s to mitigate viscous effects." The meaning of this statement is not clear. Slow pulling would tend to emphasize viscous effects, while fast pulling would emphasize elastic effects. If the slow, viscous effects are being emphasized, it is not clear how to interpret the stress/strain relationship. Some clarification of these issues is needed. Indeed, it would be interesting to know if different results are obtained at different pulling rates.

4. In Fig. 2, it is reported that endodermal cells expressing deAct-GS1 undergo less increase in cell area than controls in response to stretching force, and the researchers suggest that this may be because actin is required for shortening the cells in the apical-basal dimension, allowing them to expand in a lateral dimension. This hypothesis could be tested by fixing and sectioning deAct-GS1 and control embryos before and after stretching. Also, no mechanism is suggested as to how loss of actin would prevent shortening of the apical-basal axis in response to lateral stretching. Endoderm stiffness is not affected by deAct-GS1, so it must be something else. From the movies it looked like there may have been more cell rearrangements in deAct-GS1-treated endoderm than in controls. Could deAct-GS1-treated embryos be undergoing rearrangements in response to stretching while control cells stretch along the lines of stress? Are the cells of deAct-GS1-treated tissues less adherent to each other?

5. In the experiments with Dispase, it is reported that Dispase significantly reduces the stiffness of the endoderm while affecting to a much lower degree the coupling between the endoderm and the adjacent mesodermal tissues. There could be several reasons for this that are not discussed in the paper. First, Figure 3E shows a weakening of the fibronectin staining but not its disappearance. Thus, it could be that fibronectin is important for linkage to the mesoderm, but there is still enough fibronectin present after dispase treatment to maintain linkage but not enough to maintain endoderm stiffness. Additionally, and perhaps more important, Dispase will digest some ECM proteins more than others. It is possible (likely) that there are different ECM proteins involved in ECM stiffness (particularly those associated with the basement membrane) and others more involved with linkage between tissue layers (particularly the fibrillar components) and that these are affected differentially by Dispase. Since this is a techniques paper, it is probably not necessary to try to resolve mechanism, but the possible mechanisms should be discussed in greater depth and the authors should avoid drawing conclusions regarding mechanism that are not proven by the data.

6. The section on neural tube closure (Fig. 4) is quite preliminary and superficial. I understand that the authors want to show proof of concept that the system could be used for other epithelia. But only 5 embryos are studied. It is not clear where the neural tube is rupturing or what is the behavior of neural tube cells before and during the pulling and rupturing. Overall, I think that this section weakens rather than strengthens the paper. Either this section should be strengthened, or it should be removed in favor strengthening the mechanistic aspects of the endodermal studies.

7. Since this is a Techniques paper, it would be helpful if the authors could comment on the possible applicability of the technique to other tissues and even species.

Minor comments:

1. In the legend of Figure 2F: "Initials" should be "Initial"

2. In Line 502, I believe "2F" should be "2G"

3. Figure 3D: It was not clear how these cross sections were obtained. Are they actual sections or virtual sections?

Reviewer 3

SUMMARY OF THE ADVANCE MADE IN THIS PAPER AND ITS POTENTIAL SIGNIFICANCE TO THE FIELD

The manuscript by Oikonomou et al. (#dev.204561) describes a technique to quantitatively apply exogenous forces on the order of 1-100 μN to the endodermal epithelium in vivo. The authors report several proof-of-concept experiments that revealed fundamental insights into the mechanics of embryonic epithelia morphogenesis in vivo in the early avian embryo. We feel the reported techniques can be applied to other model systems, but this has not been discussed. This Techniques and Resources Report demonstrates novel and straightforward tools to measure and perturb morphogenesis in vivo and should be accepted with minor revisions.

SUGGESTIONS TO AUTHORS

Major points

None

Minor points

Introduction:

Please better explain to general developmental biologists the biological significance of forces in the 1-1,000 μN range.

-Lines 192-193: 'with endoderm-specific expression of a fluorescent reporter (Nerurkar et al., 2019)'.

This could easily be interpreted as endoderm-specific promoter driving expression of a fluorescent reporter instead of the authors' focally electroporating endoderm with the ubiquitously expressed pCAG-H2B-GFP to visualize endoderm cell nuclei.

-Line 362: F-actin was disrupted in a subpopulation of endoderm cells by focal electroporation to misexpress DeAct-GS1 (DeAct), a GFP-fused peptide that sequesters G-actin monomers, leading to depolymerization of the F-actin (Harterink et al., 2017).

Please explain how DeAct-GS1 works in more detail for the reader. DNA electroporations tend to be heterogeneously expressed in different cells. How do expression levels of DeAct-GS1 correlate with observed perturbations?

-Line 743: For dispase treatment, embryos were fitted with a small plastic confining ring to retain dispase solution...

What is the size of the confining ring? Can dispase be injected into a region of interest to observe more restricted perturbations?

-Line 802: Finally, stretching of the endoderm was achieved by movement of the tungsten cantilevers against filter paper squares (2 x 2 mm) placed directly on the ventral surface of the embryo.

Is it possible to use smaller filter paper squares? If yes, would the smaller filter paper permit more focused perturbations?

-Line 609: Measurement of tensile properties of the neuroepithelium during primary neurulation... The final section on quantifying the tensile forces necessary to unzip the forming neural tube is interesting but too brief. If space permits, consider expanding this section to show this approach also applies to ectoderm.

First revision

Author response to reviewers' comments

Reviewer 1

1. The introduction lacks a series of important references on using stretching devices to characterise deformations in epithelia (e.g. Harris et al. 2012; Yang et al. 2024 - Charras & Davidson labs) and deformation in whole chicken embryos (e.g. Nelemans. et al. 2020; Kunz, et al. 2023 - Smit and Xiong labs).

We thank the reviewer for raising these important references, which were omitted as an oversight. We have added them to the Introduction (lines 147-151).

2. Uniaxial stretching of epithelial tissues or full chicken embryos is not particularly novel. The novelty might be in the measurement of the applied forces using the calibrated cantilevers. However, the authors should stress this aspect more in the first study case: When the authors study the stretching of the endoderm, forces are only mentioned in Fig. S4. Instead, the authors do an in-depth characterisation of the response of endodermal cells to deformation and the role of f-actin. Although some of these results are interesting, they feel too long and out of place since do not contribute to showcase the ability of measuring forces.

We agree that the novelty of force measurement in our system could be better highlighted. Therefore, we have added endoderm force-strain measurements earlier in the manuscript (Fig. 1G, lines 207-2010). In addition, we trimmed down the discussion of cell response to deformation. However, we also note that while stretching of epithelial tissue and full chicken embryos have been reported elsewhere, the ability to apply stretch directly to an embryonic epithelium in vivo is novel, and necessary for relating tissue scale mechanics (Fig. 1) to cell-scale mechanical heterogeneity (Fig. 2), as well as isolating mechanical interactions between germ layers (Fig. 3).

3. The second (mechanical coupling of the endoderm and paraxial mesoderm) and third ("unzipping" the neural tube) cases are better representations of the strengths of the proposed method, but the third case feels incomplete in comparison to the first case where great characterisation is presented. We have now expanded the third case, extending the analysis of cell shape changes under exogenous tension to now characterize the posterior neural plate (lines 612-611 ; 662-663, Fig. 4C-G and Supplementary Fig S6). Analysis of the response of neural plate cells to tissue-scale stretch strengthens the manuscript by better illustrating broader utility of the technique, and in illustrating tissue-specific biomechanical behaviors that contrast between endoderm and ectoderm. For example, similar tissue-scale strain applied in both tissues produce markedly less cell deformation in the ectoderm vs. endoderm, despite similar tissue stiffnesses overall. This may suggest load transmission across cell-cell contacts is less efficient in the neural plate than the endoderm, an area for future investigation. In addition, we increased the number of replicates for measurement of neural tube 'unzipping' forces to better capture the range of biological variability (Fig

4B).

4. This work is presented as a technique and resources report. However, in my opinion to be used as a resource the authors should include a supplement with detailed information of how to build the device step by step with images and diagrams (including consideration such as the microscope...), how to calibrate it and possibly any code developed to compute the strain tensors.

Thank you, this is an excellent suggestion, which was shared by other reviewers as well. We have composed a supplemental assembly guide, which includes a step by step description of constructing the device, together with images, and additional detailed, practical information on how to implement the stretching device for interested labs. This assembly guide, along with all developed code for performing strain calculations, analysis of morphometrics, dimensionality reduction, clustering, etc. are all available for download on github, which is linked from the manuscript.

5. As a resource paper it would benefit for a more in-depth exploration of the sources of variability in the force measurements.

We have added several experiments to characterize sources of variability in our system. This includes further testing force-deflection linearity of cantilevers (Supplemental Fig. S1A), performing stiffness measurements at varying strain rates to confirm that viscous effects have been effectively minimized (lines 906-910, Fig. 1G), and comparing predicted vs measured cantilever base position during stage actuation (lines 902-905, Supplemental Fig. S1C). Overall, these data indicate that variability in the force measurement is far below the observed variability when quantifying mechanics of the endoderm and ectoderm. This would suggest that biological variability between samples, or possibly adherence of filter papers to the epithelium, are more likely sources of variability than the device design or viscous effects.

6. As a resource paper it would benefit from comparison with other methods to measure tissue forces. How easy are to implement? What range of forces can be measured?
- We agree that a comparison to other methods of measuring tissue forces would be ideal. However, Techniques & Resources papers at Development are subject to a restrictive word limit; we were at this limit in the original submission, and have had to cut considerably in order to accommodate substantive new experiments to address reviews in the present revision. Because we have previously discussed this topic in detail elsewhere (Loffet, Durel, and Nerurkar, Integrative & Comparative Biology 2023), we have opted to omit this discussion in the present manuscript.*

7. The mechanotypes identified in the first case of study are interesting but the biological significance and reproducibility of these UMAP clusters are not fully explored. Are they specific spatial distribution or localisations of these mechanotypes? What is the potential functional relevance of having different mechanical responses within the same epithelium? The fact that the main determinant of the cellular response to deformation is the original area, might suggest that these mechanotypes are composed of cells at different phases of the cell cycle. These could be accounted for and potentially regressed out of the data in future studies.

These are excellent questions, and we have modified the text to address them (lines 246-252). Cell position along the medio-lateral axis has no correlation with mechanotype (Supplemental Figure S2C, 3A), suggesting cells with heterogeneous mechanical properties are mixed along this axis. Antero-posterior position was not a variable in our analysis as it was kept constant. We do not yet know whether these differences are functionally important, but this is an exciting question we hope to address in future work through combining this approach with live imaging of morphogenetic cell movements. For example, in unpublished work we have identified extensive intercalary cell movements in the presumptive midgut endoderm, and it is compelling to wonder whether differing mechanical properties are necessary for these types of cell rearrangements. Due to space constraints as a Techniques paper, and how speculative this point is, we have only briefly noted that it is presently unclear whether the mechanical heterogeneity observed is functionally important or just a secondary consequence of cell type diversity in a more conventional sense, or alternatively important for enabling cell rearrangements and limiting cell jamming during

epithelial morphogenesis.

8. What is the source of mechanical coupling between the endoderm and paraxial mesoderm? Does this coupling have a functional role?

This is an important question that arises from our work, but one that we cannot yet fully answer. Prior work from Sato, Lansford and colleagues strongly suggested that fibronectin pillars form the mechanical connection between endoderm and somites, and so we were surprised to see that dispase treatment did not have a more marked effect on strain transfer, despite dramatically disrupting the fibronectin matrix between these layers. It is possible that active cell-cell interactions across the interface, or ECM components that are less sensitive to dispase digestion, could explain this coupling, and we have added text to note this in the manuscript (lines 566-570). Functionally, the tight coupling between germ layers is likely important both for mechanical integrity/stability of the developing embryo, and potentially for constraining or coordinating complex morphogenetic behaviors across the endoderm and mesoderm. We originally did include a discussion of one putative example in the form of maintaining bilateral symmetry during axis elongation and somitogenesis, events related to scoliosis that have been viewed mechanically as isolated events in the mesoderm, but may need to be reconsidered under the boundary constraint of a stiff endodermal neighbor.

Reviewer 2

1. This is a Techniques and Resources paper. Thus it is important that the Methods be described sufficiently so that others could construct the apparatus in their own labs. It would be very helpful if it could be better described how the tungsten cantilevers are attached to the filter paper, as this was not clear from the description in the paper. It would also be helpful to include a technical sketch of the system.

We have written a detailed supplemental methods section that serves as a 'how to' assembly guide for building and implementing the stretcher device, including many pictures and a more detailed description, which should improve the ability of readers to implement the device on their own (lines 970-973). This assembly guide, along with all developed code for performing strain calculations, analysis of morphometrics, dimensionality reduction, clustering, etc. are all available for download on github, which is linked from the manuscript.

2. Does the total area of the filter paper and its area of contact with the embryo affect the measurements being made?

Thank you for raising this important point, which was unclear in our original submission. The contact area will likely not change the overall behaviors observed (e.g. cell area plateau, effects of actin disruption, endoderm-mesoderm strain transfer), but it will change the force magnitude associated with these behaviors. A larger contact area would produce a lower stress (force per unit area) for a given force, diluting the effects. A smaller contact area would increase the force and stress concentrations, and likely produce more heterogeneous strain fields. We have added a brief discussion of the contact area effects to the methods (lines 882-893). See response to Reviewer 3, #5, below as well.

3. It is stated: "The tungsten cantilevers were displaced slowly at a rate of 0.003 mm/s to mitigate viscous effects." The meaning of this statement is not clear. Slow pulling would tend to emphasize viscous effects, while fast pulling would emphasize elastic effects. If the slow, viscous effects are being emphasized, it is not clear how to interpret the stress/strain relationship. Some clarification of these issues is needed. Indeed, it would be interesting to know if different results are obtained at different pulling rates.

To briefly explain, soft tissues are viscoelastic, exhibiting behaviors of both solids and fluids. The stress in a fluid depends on the rate of deformation: faster deformation rate is associated with higher stress, as when one experiences greater resistance if quickly pulling a knife from a jar of honey vs. doing so very slowly. On the other hand, stress in a solid depends on the magnitude of deformation, not the rate (stretching a rubber

band quickly vs. slowly will require the same force). When one deforms a viscoelastic material quickly, the rate-dependent (fluid) behaviors are more pronounced. For this reason, it is convention in soft tissue biomechanics to minimize viscous effects by using a very slow deformation rate. The stress associated with fluid flow as the solid components of the tissue deforms are minimized by the slow rate of deformation, as water in the tissue has time to flow along local pressure gradients until they equilibrate. In order to illustrate this and validate the testing protocol, we added new data comparing different displacement rates (lines 906-910, Fig. 1G). We find across the strain rates compared that there is no significant difference in measured properties. Increasing the displacement rate by 2-fold does trend toward higher effective stiffness, while 2-fold slower has no discernable effect. This supports the notion that 3 μ m/s is sufficiently slow as to minimize viscous effects.

4. In Fig. 2, it is reported that endodermal cells expressing deAct-GS1 undergo less increase in cell area than controls in response to stretching force, and the researchers suggest that this may be because actin is required for shortening the cells in the apical-basal dimension, allowing them to expand in a lateral dimension. This hypothesis could be tested by fixing and sectioning deAct-GS1 and control embryos before and after stretching. Also, no mechanism is suggested as to how loss of actin would prevent shortening of the apical-basal axis in response to lateral stretching. Endoderm stiffness is not affected by deAct-GS1, so it must be something else. From the movies it looked like there may have been more cell rearrangements in deAct-GS1-treated endoderm than in controls.

Could deAct-GS1-treated embryos be undergoing rearrangements in response to stretching while control cells stretch along the lines of stress? Are the cells of deAct-GS1-treated tissues less adherent to each other?

These are excellent suggestions. As requested, we have now fixed embryos before and after stretching to image cell height. These data confirm that deAct-GS1 expressing cells have significantly greater cell height than control myr-RFP expressing cells within the same embryo when subject to tensile stretch (line 507-509, Supplemental Fig. S4E-G). As far as the mechanism for actin-dependent apico-basal shortening, we can only speculate within the scope of the study, and have added a brief discussion of how loss of cytoskeletal connectivity within the epithelium may limit out-of-plane (Poisson's) force transmission (line 509-513). Due to strict space constraints of the Techniques format, we are precluded from an extended discussion, but do hope to further investigate this in future work. As far as cell rearrangements, stretching is performed progressively on a time scale of 5-10 minutes. During this time, we do not observe obvious active cell rearrangements, but instead cell displacements that are affine/consistent with the broader applied deformation to the tissue. Therefore, we do not expect that cell rearrangements come into play just yet. However, in ongoing work for a future publication, we are adapting this system to perform live imaging and observe influences of exogenous tension on cell rearrangements over larger time scales that are more relevant in the context of gut tube formation (hours). We hope to gain more insights into this topic through those ongoing and future studies.

5. In the experiments with Dispase, it is reported that Dispase significantly reduces the stiffness of the endoderm while affecting to a much lower degree the coupling between the endoderm and the adjacent mesodermal tissues. There could be several reasons for this that are not discussed in the paper. First, Figure 3E shows a weakening of the fibronectin staining but not its disappearance. Thus, it could be that fibronectin is important for linkage to the mesoderm, but there is still enough fibronectin present after dispase treatment to maintain linkage but not enough to maintain endoderm stiffness. Additionally, and perhaps more important, Dispase will digest some ECM proteins more than others. It is possible (likely) that there are different ECM proteins involved in ECM stiffness (particularly those associated with the basement membrane) and others more involved with linkage between tissue layers (particularly the fibrillar components) and that these are affected differentially by Dispase. Since this is a techniques paper, it is probably not necessary to try to resolve the mechanism, but the possible mechanisms should be discussed in greater depth and the authors should avoid drawing conclusions regarding mechanism that are not proven by the data.

We agree that our initial interpretation was an oversimplification, and have added statements to reflect that disperse treatment weakens but does not eliminate fibronectin at the interface, and that it is possible that disperse-sensitive ECM, rather than collagen and/or elastin explicitly, may play a greater role in determining in plane properties than interfacial properties between germ layers (lines 566-570).

6. **The section on neural tube closure (Fig. 4) is quite preliminary and superficial. I understand that the authors want to show proof of concept that the system could be used for other epithelia. But only 5 embryos are studied. It is not clear where the neural tube is rupturing or what is the behavior of neural tube cells before and during the pulling and rupturing. Overall, I think that this section weakens rather than strengthens the paper. Either this section should be strengthened, or it should be removed in favor of strengthening the mechanistic aspects of the endodermal studies.**
We have expanded on this considerably in the revision, following the recommendation of all three reviewers to do so, and general enthusiasm for growing this application from Reviewers 1 and 3. First, we increased the number of replicates for neural tube unzipping experiments to better characterize the biological variability in dorsal rupture of the tube under tension (Fig. 4A, B). Next, we performed measurements of cell morphometrics in the neural plate as a function of applied stretch, similar to the endoderm experiments (lines 612-611 ; 662-663, Fig. 4C-G). These new data offer an insightful comparison between biomechanics of endoderm and ectoderm derived epithelia, which share some behaviors but are overall quite distinct. See also response to Reviewer 1 #3 above.
7. **Since this is a Techniques paper, it would be helpful if the authors could comment on the possible applicability of the technique to other tissues and even species.**
We have added a discussion (lines 910-934) of the applicability to other tissues and species, including estimates of the force magnitudes in those systems. The approach can also be extended to studying tissue explants and organoids, which is now indicated as well.
8. **In the legend of Figure 2F: "Initials" should be "Initial"**
Thank you for catching this, we have corrected it.
9. **In Line 502, I believe "2F" should be "2G"**
Thank you, we have corrected this.
10. **Figure 3D: It was not clear how these cross sections were obtained. Are they actual sections or virtual sections?**
We have clarified the text in the figure legend - these are whole mount thick slabs of the embryo cut transversely and viewed end-on (line 653).

Reviewer 3

1. **Please better explain to general developmental biologists the biological significance of forces in the 1-1,000 μN range.**
*To add context, we have added that this equates to the weight of between 100 nL and 100 μL of water, respectively (line 183). We have also added a brief discussion to the Methods on the broad applicability of this approach, including other morphogenetic processes where estimated force magnitudes are within this range, such as avian axis elongation and *Xenopus* gastrulation, among others (lines 910-934).*
2. **Lines 192-193: 'with endoderm-specific expression of a fluorescent reporter (Nerurkar et al., 2019)'. This could easily be interpreted as endoderm-specific promoter driving expression of a fluorescent reporter instead of the authors' focally electroporating endoderm with the ubiquitously expressed pCAG-H2B-GFP to visualize endoderm cell nuclei.**
We agree the suggested wording is clearer and have modified the sentence accordingly (lines 193-195).

3. **Line 362: F-actin was disrupted in a subpopulation of endoderm cells by focal electroporation to misexpress DeAct-GS1 (DeAct), a GFP-fused peptide that sequesters G-actin monomers, leading to depolymerization of the F-actin (Harterink et al., 2017). Please explain how DeAct-GS1 works in more detail for the reader. DNA electroporations tend to be heterogeneously expressed in different cells. How do expression levels of DeAct-GS1 correlate with observed perturbations?**

We have added details to the methods that further explain the mode of action for GS-1 in its disruption of F-actin in cells (lines 732-738). Regarding dosage effects related to electroporation heterogeneity, qualitatively we observed only a weak correlation between GFP intensity and cell phenotype, and only at low expression levels. However, moderate to high expression produced equally robust effects on the actin cytoskeleton, consistent with passing a threshold that destabilizes the equilibrium between monomeric and filamentous actin and leads to catastrophic disassembly of the cytoskeleton. This behavior is also observed when G-actin is sequestered pharmacologically using Latrunculin A or similar drugs (e.g. Ayscough et al. JCB 1997). Harterink et al. compared low and high DeAct-GS1 expression in vitro, finding that high expression inhibited cell migration, while lower expression levels had no effect in this regard, further suggesting it functions more as a switch than a dosage dependent perturbation.

4. **Line 743: For dispase treatment, embryos were fitted with a small plastic confining ring to retain dispase solution. What is the size of the confining ring? Can dispase be injected into a region of interest to observe more restricted perturbations?**

Confining rings are 1 cm in diameter, this detail has been added to the methods (line 798). Regarding localized disruption via dispase injection, this has been recently carried out in the chick embryo by Tom Schultheiss' group to study elastic energy storage in the lateral plate mesoderm of chick embryos (Zaher et al. Cell Reports 2025). While injected locally, their results suggest that ECM is degraded the left half of the embryo when injected into the coelomic cavity separating the somatopleure and splanchnopleure. Therefore targeted regional disruption, particularly in the endoderm which lacks a localized coelomic space into which injections can be performed, may be challenging, requiring use of hydrogels or other slow, localized release mechanisms.

5. **Line 802: Finally, stretching of the endoderm was achieved by movement of the tungsten cantilevers against filter paper squares (2 x 2 mm) placed directly on the ventral surface of the embryo. Is it possible to use smaller filter paper squares? If yes, would the smaller filter paper permit more focused perturbations?**

We have added a brief discussion of filter paper size effects to the methods (lines 882-893), indicating that, as posed by the reviewer, smaller filter paper can be used to produce more focused perturbations. We chose this particular size for the present study in order to ensure a uniform stress field in the region of interest, so that we could confirm that heterogeneity in the cell responses to stretch was due to differences in cell mechanics rather than heterogeneity in the forces applied. As the filter used becomes smaller, the cells being analyzed are closer to the free anterior and posterior ends of the filter paper, beyond which the stress and strain will decay toward endogenous/residual levels. As a result, 'boundary effects' come into play within the region of interest, greatly complicating interpretation of results. A second consideration is the magnitude of forces one can reliably detect, as the smaller the filter paper, the smaller the forces must be applied to avoid excessive stress concentrations. In other words, the same force applied by the cantilever is more prone to tear the tissue in a smaller filter paper than a larger one. Sensitivity of the force measurement can be tuned: bending stiffness varies with cantilever radius $\sim R^4$ and with the inverse of cantilever length L^3 , such that a thinner, longer cantilever will have much greater sensitivity for force measurement and application. See also response to Reviewer 2, #2 above.

6. **Line 609: Measurement of tensile properties of the neuroepithelium during primary neurulation... The final section on quantifying the tensile forces necessary to unzip the forming neural tube is interesting but too brief. If space permits, consider expanding this**

section to show this approach also applies to ectoderm.

We have expanded this section through additional experiments, which now includes a full biomechanical analysis of neural plate cells (lines 612-611 ; 662-663, Fig 4C-G and Supplemental Fig. S6). These new data not only further illustrate the applicability of the approach to tissues beyond the endoderm, they also highlight tissue-specific differences between epithelia, as the ectoderm appears to rely more on intercellular deformations to accommodate tissue strain, with cells deforming significantly less than endoderm cells. See also Reviewer 1 #3 above.

Second decision letter

MS ID#: dev.204561R1

MS TITLE: Application of tissue-scale tension to avian epithelia in vivo to study multiscale mechanics and inter-germ layer coupling

AUTHORS: Panagiotis Oikonomou; Lisa Calvary; Helena Campos Cirne; Andreas Emerson Welch; John Franics Durel; Olivia Powell; Kwantae Kim; Nandan Nerurkar

Dear Dr Nerurkar,

I have now received all the referees reports on the above manuscript, and have reached a decision. The referees' comments are appended below, or you can access them online: please go to .

The overall evaluation is positive and we would like to publish a revised manuscript in Development, provided that the referees' comments can be satisfactorily addressed. Please attend to all of the reviewers' comments in your revised manuscript and detail them in your point-by-point response. In particular, rev 1 raises an excellent point in that the quality of the supplementary methods explaining how to make and use this device (GitHub) are still really limited. Since this is a techniques paper, it is essential that the level of detail is sufficient for others to adopt the technique. If you do not agree with any of their criticisms or suggestions explain clearly why this is so. If it would be helpful, you are welcome to contact us to discuss your revision in greater detail. Please send us a point-by-point response indicating your plans for addressing the referees' comments, and we will look over this and provide further guidance.

Reviewer 1**SUMMARY OF THE ADVANCE MADE IN THIS PAPER AND ITS POTENTIAL SIGNIFICANCE TO THE FIELD**

The paper is better shape, the narrative is clearer and my questions have been answer in a great degree. Nevertheless I still have some suggestions:

Important:

Since this is resources paper I would expand the instructions at GitHub. At the moment they are very limited, only showing the assembly of the device, probably not enough to be used. The authors could show:

- > 1. how to couple the device to the microscope
- > 2. How to mount the embryo in the device
- > 3. How to extract the data from the device
- > 4. Any code to analyse the results and compute the forces. I know this the authors included some analysis image analysis code, but that can be found elsewhere. In my view the most important is the code used for measurements and calibration.

Other minor comments:

> The title and summary statements only mention the ability of applying tension, while the novelty part, as I understand it comes from the ability of measuring forces. This should be reflected in the titles and summary statement.

> The authors did a review comparing different methods in Loffet, Durel, and Nerurkar, Integrative & Comparative Biology 2023. Although they cite this paper, it is mentioned in a manner that indicates that this a resource of that nature, and they should indicate it explicitly in the text.

Reviewer 2

SUMMARY OF THE ADVANCE MADE IN THIS PAPER AND ITS POTENTIAL SIGNIFICANCE TO THE FIELD

The authors have responded substantively to points that I raised. I am not sure, however, how generally applicable the methods would be to other systems. The modifications that would be necessary would be so substantial that it would really amount to a separate technique. As the authors point out, the current technique depends on broad attachment of the tension apparatus to the embryo via filter paper. It is not clear how/if an apparatus that accomplishes this could be developed in other experimental systems. In addition, since the filter paper is attached to the external surface of the embryo, it could not be used to probe internal, mesodermally-derived tissues. These considerations decrease the general applicability of the methods. The technique is very nice and useful for a specific set of research questions in a specific experimental system. In this light, I think that it may be more appropriate to submit a traditional research article which reports on new biological insights generated using these methods, as opposed to a Methods paper. Such a Research Paper would require further experiments, since the current study does not generate sufficient insight into mechanism.

Second revision

Author response to reviewers' comments

We thank the reviewers for their feedback, and were happy to see that overall, issues raised in the original round of reviews were satisfactorily addressed. We have made further changes for the second round of minor revisions that are indicated below.

Reviewer 1:

1. **Important: Since this is a resources paper I would expand the instructions at GitHub. At the moment they are very limited, only showing the assembly of the device, probably not enough to be used.**

We agree that more detail was needed, and have now greatly expanded the Github resource. In addition to providing greater detail on assembly of the tensile tester, we have added new sections on interfacing the device with a microscope, operating the linear actuators, mounting the embryo, collecting data, and analyzing it using code that is also available for download on GitHub. In doing so, the focus of this document has broadened from device assembly to practical usage of the device to collect data, and how to analyze it. This is an important point, and we thank the reviewer for raising it.

- a. **how to couple the device to the microscope**

As described in the revised Github document, the device simply sits atop the microscope stage. The weight of the base plate is sufficient to maintain a stable foundation for experiments to be performed directly on top of the base while it rests on the microscope stage.

- b. **How to mount the embryo in the device**

As described in greater detail in the revised Github document, the embryo remains on filter paper rings used during harvest from the egg and subsequent electroporation. The embryo is placed atop a 35 mm culture dish filled to the top

with semisolid EC culture medium. The dish itself is placed on the base of the device during the stretching experiments.

c. How to extract the data from the device

We have added this to the revised Github resource. Raw data is images, collected on the microscope PC using the image acquisition software. In our case, Zeiss' Zen software is used to generate .dzi files. Frame number, actuator velocity, and observed displacement of the cantilever tip in sequential images, are sufficient for quantification of applied force from these data. This is now more clearly indicated in the document.

d. Any code to analyse the results and compute the forces. I know this the authors included some analysis image analysis code, but that can be found elsewhere. In my view the most important is the code used for measurements and calibration. We have also added greater detail to describe how this is done, including directly mentioning each specific function/program as part of the analysis pipeline. All custom code is available on Github, and the document now clearly indicates how each is used in sequence to process and visualize data.

2. Title and summary statements only mention the ability of applying tension, while the novelty part, as I understand it comes from the ability of measuring forces. This should be reflected in the titles and summary statement.

In the title, we have changed "Application of tension" to "Application and measurement of tension". We have made similar changes to the summary statement.

3. The authors did a review comparing different methods in Loffet, Durel, and Nerurkar, Integrative & Comparative Biology 2023. Although they cite this paper, it is mentioned in a manner that indicates that this a resource of that nature, and they should indicate it explicitly in the text.

We have modified the text to make clear that this reference is a review of the topic rather than original research on one particular method.

Reviewer 2:

The authors have responded substantively to points that I raised. I am not sure, however, how generally applicable the methods would be to other systems. The modifications that would be necessary would be so substantial that it would really amount to a separate technique. As the authors point out, the current technique depends on broad attachment of the tension apparatus to the embryo via filter paper. It is not clear how/if an apparatus that accomplishes this could be developed in other experimental systems. In addition, since the filter paper is attached to the external surface of the embryo, it could not be used to probe internal, mesodermally-derived tissues. These considerations decrease the general applicability of the methods. The technique is very nice and useful for a specific set of research questions in a specific experimental system. In this light, I think that it may be more appropriate to submit a traditional research article which reports on new biological insights generated using these methods, as opposed to a Methods paper. Such a Research Paper would require further experiments, since the current study does not generate sufficient insight into mechanism.

We thank the reviewer for their feedback, including many insightful comments in our first submission that greatly strengthened the manuscript and, according to the reviewer, we have addressed. The format as a Techniques and Resources submission is something we gave great thought to before submitting the Presubmission Inquiry to the Editorial Board, and again once we were invited to submit the full manuscript in this format. Our reasoning for this format is detailed below.

First, it is common for Techniques & Resources papers in Development to restrict their focus to a single model organism. There are many such examples in just the last 3 months, including knock-in of DNA into zebrafish (Oikemus et al. 2025), manipulating and measuring localized translation of erm-1 in C elegans (van der Salm et al. 2025), chromatin profiling in Hofstenia miami (Bump et al. 2025), relating cell proliferation to tissue morphology in mice (Lo Vercio et al. 2025). We also note the recent publication of a highly complementary work to our own, by Pourquie, Mahadevan, and colleagues on the development of a technique to measure axis

elongation forces in the chick embryo, which appeared as a *Techniques and Resources* paper as well (Chan et al 2023).

Second, on general applicability of the method, aside from the filter paper attachment, every other aspect of the setup can be seamlessly applied to other models, including strain quantification, tissue stiffness measurements, cell morphometrics, UMAP, etc. As for the filter paper attachment itself, it is likely this could be applied in early amphibian embryos to study gastrulation or neurulation, though testing this directly is beyond the scope of the present work. Moreover, when one considers compressive forces, no filter paper is needed and translatability to other tissues/models should be quite feasible.

Third, at minimum, we demonstrate here that the method can be applied to chick endoderm and ectoderm. It would be trivial to extend this approach to the avian epiblast. From the conceptual standpoint, we demonstrated that this method gives one access to study mechanical coupling between germ layers, mechanical heterogeneity of epithelial cells, and tissue-scale mechanical properties of epithelial tissues. These are likely to be important considerations for studying the mechanics of fundamental events in embryonic development that are of broad interest, including gastrulation, epiboly, neurulation, neural tube closure, and gut tube formation.

Fourth, as discussed in the Introduction, many methods already exist to probe the mechanics of mesodermally-derived tissues, while measuring mechanics of embryonic epithelia in vivo has remained particularly challenging. Extending from the latter to the former would not provide a meaningful new contribution to the field when many such techniques already exist.

In sum, we respectfully disagree with Reviewer 2 on this point, and maintain that the present work is most appropriate as a *Techniques & Resources* pape. This is based on wide precedent for publishing works of similar scope in this format, translatability of the approach to other systems, and the fundamental nature of questions the method gives access to in the chick embryo.

Third decision letter

MS ID#: dev.204561R2

MS TITLE: Application and measurement of tissue-scale tension to avian epithelia in vivo to study multiscale mechanics and inter-germ layer coupling

AUTHORS: Panagiotis Oikonomou; Lisa Calvary; Helena Campos Cirne; Andreas Emerson Welch; John Franics Durel; Olivia Powell; Kwantae Kim; Nandan Nerurkar

Dear Dr Nerurkar,

I am happy to tell you that your manuscript has been accepted for publication in *Development*, pending our standard publication integrity checks.